# Evaluation of transcutaneous electrical acupoint stimulation for improving pain and cognitive function in elderly patients around the perioperative period of hip replacement surgery: A meta-analysis

**Sujuan Xu[1]◉, Kai Huang[2]◉, Qing Jiang(ID)[3]\***

**1** Comprehensive Internal Medicine Department of Zhijiang Campus, Tongde Hospital of Zhejiang Province, Hangzhou, China, **2** Department of Orthopedics, Tongde Hospital of Zhejiang Province, Hangzhou, China, **3** Department of Anesthesiology, Tongde Hospital of Zhejiang Province, Hangzhou, Zhejiang, China

◉ These authors contributed equally to this work.

\* 13857125431@163.com

**Data Availability Statement:** All relevant data are within the manuscript file.

## Abstract

### Purpose

We aim to evaluate the efficacy and safety of transcutaneous electrical acupoint stimulation (TEAS) in elderly patients around the perioperative period of hip replacement surgery.

### Methods

The China National Knowledge Infrastructure(CNKI), Wangfang Data, VIP database, SinoMed, PubMed, and Embase databases were searched for relevant publications until August 2024. All randomized controlled studies evaluating the efficacy and safety of TEAS in patients around the perioperative period of hip replacement surgery. We calculated pooled risk ratio (RR) with 95% CIs for binary outcomes and standardized mean difference (SMD) for continuous outcomes. The Cochrane's Risk of Bias Tool were used to evaluate the quality of studies.

### Results

A total of 13 studies with 946 patients were included in this analysis. 1-day visual analogue scale (VAS) scores and 2-day VAS scores were significantly lower in the TEAS group compared to the control group (SMD: -0.78, 95% CI: -1.47, -0.09, $P = 0.02$ and SMD:-0.54, 95% CI:-1.00,-0.09,$P = 0.02$). Furthermore, 1-day mini-mental state examination (MMSE) scores and 3-day MMSE scores were significantly higher in the TEAS group compared to the control group (SMD: 1.60, 95% CI: 0.68, 2.51,$P$<0.001 and SMD:1.31, 95% CI:1.03,1.59, $P$<0.001), along with a lower postoperative cognitive dysfunction rate (RR: 0.55, 95% CI: 0.41, 0.73, $P$<0.001).

**Funding:** Zhejiang Province Traditional Chinese Medicine Science and Technology Plan Project (NO.2024ZL029).

**Competing interests:** The authors have declared that no competing interests exist.

## Conclusions

Our meta-analysis demonstrated that TEAS significantly reduces pain and improves cognitive function in patients undergoing hip replacement surgery. Future studies should further investigate the optimal TEAS protocols to maximize these benefits across different population and surgical settings.

## 1. Introduction

Hip replacement surgery has become a widely accepted procedure for alleviating pain and improving the quality of life in patients with severe joint diseases [1]. Despite advancements in surgical techniques that have enhanced its success, postoperative recovery remains a significant challenge [2]. A considerable proportion of patients undergoing hip replacement—between 16% to 45%—experience postoperative cognitive dysfunction (POCD), and more than 60% suffer from severe pain following the operation [3–5]. These complications highlight the importance of developing effective strategies to improve pain management and cognitive function during the perioperative period, ultimately enhancing elderly patient outcomes and recovery quality.

Traditionally, the management of postoperative pain and the prevention of cognitive impairment have relied heavily on drug-based therapies [6]. Yet, these methods come with their set of challenges, including adverse reactions, possible drug interactions, and the risk of dependency or tolerance, particularly with opioid pain relievers [7]. Recognizing this, The American Pain Society advocates for a patient- and procedure-specific plan for effective postoperative pain management [8]. Emphasizing a multimodal strategy, these guidelines suggest incorporating various treatment modalities to address pain comprehensively, enhancing safety and efficacy in postoperative recovery [8].

Transcutaneous electrical acupoint stimulation (TEAS) emerges as a promising non-invasive electrostimulation technique with multiple advantages including painlessness, standardized parameters, and ease of use. Recognizing its potential, the American Pain Society, along with partners, has recommended clinicians consider TEAS as an adjunct to other postoperative pain treatments [8]. This endorsement highlights TEAS's growing acceptance in clinical practice as an innovative approach to enhancing patient care. However, this view is not universally held, the Royal College of Surgeons of England and the College of Anesthetists have concluded that TEAS should not be offered for acute postoperative pain, citing a lack of systematic discussion and meta-analytical evidence to robustly support its efficacy and safety [9, 10]. This controversy underscores the need for comprehensive and systematic evaluation of TEAS in the context of hip replacement surgery.

Therefore, this meta-analysis aims to evaluate the efficacy and safety of TEAS in elderly patients during the perioperative period of hip replacement surgery.

## 2. Material and methods

The meta-analysis was carried out according to the Preferred Reporting Items for a Systematic Review and Meta-analysis (PRISMA) 2020 guidelines [11, 12]. This is a systematic review and meta-analysis, ethics approval and consent to participate are not applicable.

### 2.1 Search strategy

The China National Knowledge Infrastructure(CNKI), Wangfang Data, VIP database, SinoMed, PubMed, and Embase databases were searched for relevant publications until

**Table 1. Search strategy in PubMed, Embase, Wangfang Data, China National Knowledge Infrastructure (CNKI) database, China Science and Technology Journal Database (VIP) database and SinoMed databases.**

| Database | Search strategy |
|---|---|
| PubMed (11) | ("Transcutaneous Electrical Nerve Stimulation" [Title/Abstract] OR "TENS" [Title/Abstract]) OR ("Transcutaneous Electrical Acupoint Stimulation" [Title/Abstract] OR "TEAS" [Title/Abstract]) AND ("Arthroplasty, Replacement, Hip" [Mesh] OR "hip replacement" [Title/Abstract]) |
| Embase (48) | ('transcutaneous electrical nerve stimulation'/exp OR 'TENS' OR 'transcutaneous electrical acupoint stimulation':ab,ti OR 'TEAS') AND ('hip replacement'/exp OR 'hip arthroplasty') |
| CNKI (50) | (TKA% = 经皮穴位电刺激 OR TKA% = TEAS OR TKA% = 经皮神经电刺激 OR TKA% = TENS) AND TKA% = 髋关节置换术 |
| Wangfang Data (22) | (经皮穴位电刺激 OR TEAS OR 经皮神经电刺激 OR TENS) AND 髋关节置换术 |
| VIP Database (28) | M = (经皮穴位电刺激 OR TEAS OR 经皮神经电刺激 OR TENS) AND M = (髋关节置换术) |
| SinoMed (38) | (("髋关节置换术"[常用字段:智能]) AND (("经皮穴位电刺激"[常用字段:智能] OR "TEAS"[常用字段:智能] OR "经皮神经电刺激"[常用字段:智能] OR "TENS"[常用字段:智能]))) |

August 2024. The keywords used for the search were ("Transcutaneous Electrical Nerve Stimulation" [Title/Abstract] OR "TENS" [Title/Abstract]) OR ("Transcutaneous Electrical Acupoint Stimulation" [Title/Abstract] OR "TEAS" [Title/Abstract]) AND ("Arthroplasty, Replacement, Hip" [Mesh] OR "hip replacement" [Title/Abstract]). The detail search strategy was shown in Table 1. In addition, the reference lists of the selected papers were manually searched to retrieve additional relevant articles.

## 2.2 Inclusion and exclusion criteria

Two investigators independently evaluated the eligibility of each study based on the following inclusion criteria: Population (P): elderly patients undergoing elective hip replacement surgery; Intervention (I):TEAS; Comparison (C):blank control or sham stimulation; Outcomes (O): pain levels as measured by the 1-day and 2-day visual analogue scale(VAS) scores, cognitive function assessed through the 1-day and 3-day mini-mental state examination (MMSE) scores; postoperative cognitive dysfunction (POCD) rate, and adverse events including nausea, vomiting, pruritus, and dizziness. Study design (S): randomized controlled trials. We excluded the following studies: duplicated articles, abstracts without full texts, editorial comments, letters, case reports, reviews, meta-analyses, irrelevant titles and abstracts, unpublished study protocols or clinical registrations, animal studies, and studies with irrelevant outcome measures or incomplete data.

## 2.3 Retrieval of relevant articles

Two researchers simultaneously conducted literature screening, employing both software and manual methods to eliminate duplicates. They reviewed titles and abstracts to discard irrelevant studies, followed by a full-text reading to further exclude studies that did not match the research criteria. Cross-verification was performed on the screening results. In case of differing opinions, a third author was sought for assistance.

## 2.4 Quality assessment

For randomized control trials, the quality was meticulously assessed by two independent reviewers employing the Cochrane Risk of Bias Tool [13], ensuring an objective evaluation. This detailed analysis spanned seven critical domains: the generation of random sequences to

ensure unpredictability in group assignments; the concealment of allocation to prevent selection bias; the blinding of participants and personnel to minimize performance bias; the blinding of outcome assessment to avoid detection bias; the completeness of outcome data to counteract attrition bias; the selectivity of reporting to confront reporting bias; and an exploration of other potential biases that might skew the results. Each domain's evaluation resulted in a categorization as presenting a high, unclear, or low risk of bias. This structured approach underscores the commitment to uphold the integrity and reliability of the findings presented, serving as a cornerstone for the soundness of the study's conclusions.

### 2.5 Data extraction

Two researchers(SX and KH) independently conducted data extraction for all included articles. The extracted data included the author, year, study characteristics (country, study design, outcome, comparison), patient characteristics (age, female/male, number of patients, type of anesthesia). Disagreements between the researchers were resolved via discussion until a consensus was reached. Authors were contacted for missing outcome data when it was indicated that one of the outcomes was measured but not reported.

### 2.6 Outcome measures

Pain levels are evaluated on the 1st and 2nd postoperative days using the VAS scores, where higher scores indicate more severe pain [14]. Cognitive function is assessed on the 1st and 3rd postoperative days using the MMSE scores, with higher scores denoting better cognitive function [15]. The rate of POCD serves as an indicator of cognitive decline post-surgery, contributing to our understanding of surgery's potential impact on cognitive health [16]. Additionally, the occurrence of adverse reactions, including nausea, vomiting, pruritus, and dizziness, is carefully documented as crucial indicators of the treatment's safety profile. Through these comprehensive outcome measures, we aim to thoroughly evaluate the efficacy and safety of TEAS in improving pain and cognitive function for elderly patients in the perioperative period of hip replacement surgery.

### 2.7 Statistical analysis

For continuous outcomes, we determined pooled effect estimates by calculating the standardized mean difference (SMD), and for binary outcomes, we used the risk ratio (RR), each accompanied by their respective 95% confidence intervals (CIs). To evaluate heterogeneity within and between groups, we employed the Cochrane Q and $I^2$ statistics [17]. If studies exhibit significant heterogeneity ($I^2 \geq 50\%$), a random-effects model would be utilized. Due to the limited number of included studies (fewer than 10), a meta-regression analysis to identify potential sources of heterogeneity was not feasible. Conversely, in cases of low heterogeneity ($I^2 < 50\%$), a fixed-effects meta-analysis would be conducted for comparison.

Publication bias was evaluated using a funnel plot and Egger's test. For all statistical tests, a *P*-value below 0.05 was deemed statistically significant. Statistical analyses were performed using the R version 4.3.1.

## 3. Results

### 3.1 Literature search and study selection

The initial search identified 197 publications. Among these, 60 were found to be duplicates, and an additional 116 were excluded due to not meeting the eligibility criteria. After a detailed examination of the full texts of the remaining 21 articles, 8 studies were further excluded for

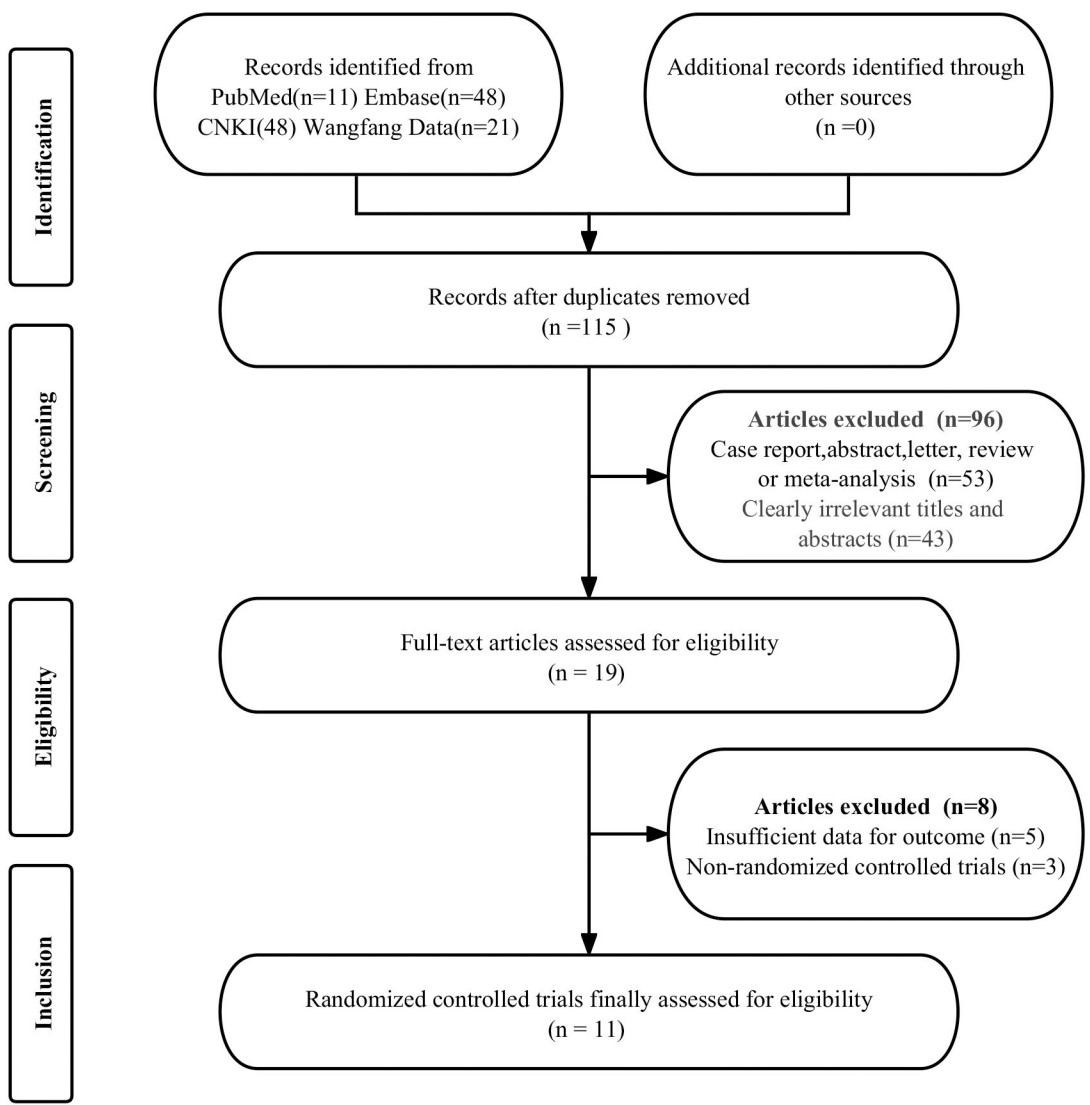

**Fig 1. PRISMA flow diagram illustrating the study selection process.**

reasons including insufficient data on outcomes (n = 5) and the studies being non-randomized control trials (n = 3). Consequently, 13 articles that assessed the efficacy and safety of TEAS in enhancing pain and cognitive function in elderly patients during the perioperative period of hip replacement surgery were selected for inclusion in the meta-analysis [18–30]. The selection process is depicted in the PRISMA flow diagram presented in Fig 1.

### 3.2 Study description and quality assessment

The 13 eligible randomized controlled studies included a total of 946 elderly patients. The number of patients in each study ranged from 40 to 120. The study and patient characteristics are summarized in Table 2.

The utilization of the Cochrane Risk of Bias Tool revealed insights concerning biases in the analyzed studies, as depicted in Fig 2. A significant risk of bias was predominantly identified in two specific domains: the blinding of participants and personnel, and the blinding of outcome

Table 2. The study characteristics of the included studies.

| Author | Year | Country | Study design | Outcome | Comparison | Mean age±SD | Male/ Female | Number of patients | Locations of electrodes placements | Duration of operation | Frequency |
|---|---|---|---|---|---|---|---|---|---|---|---|
| Sun K et al. | 2018 | China | Randomized controlled trial | (3)(4)(5) | TEAS | 86.3 ±4.4 | 7/13 | 20 | Baihui (DU20), Fengchi (GB20) | From 30 minutes pre-surgery to its conclusion | 2–100 Hz |
| | | | | | Sham stimulation | 85.8 ±4.2 | 8/12 | 20 | Baihui (DU20), Fengchi (GB20) | From 30 minutes pre-surgery to its conclusion | None |
| Yang et al. | 2023 | China | Randomized controlled trial | (3)(6)(7) (9) | TEAS | 75.36 ±2.47 | 40/20 | 60 | Hegu (LI4) and Neiguan (PC6) | From 30 minutes pre-surgery to its conclusion | NA |
| | | | | | No stimulation | 74.31 ±2.51 | 39/21 | 60 | None | None | None |
| Lan et al. | 2017 | China | Randomized controlled trial | (1)(2)(6) (7)(8)(9) | TEAS | >60 | NA | 30 | Neiguan (PC6), Hegu (LI4),Fengchi(GB20), Zusanli(ST36) | 30 minutes | 2–100 Hz |
| | | | | | Sham stimulation | >60 | NA | 30 | Neiguan (PC6), Hegu (LI4),Fengchi(GB20), Zusanli(ST36) | 30 minutes | None |
| Sun PH et al. | 2019 | China | Randomized controlled trial | (1)(2)(6) (7)(8) | TEAS | 69. 2 ±3.0 | 14/26 | 40 | Neiguan (PC6), Hegu (LI4) | From 30 minutes pre-surgery to its conclusion | 2–100 Hz |
| | | | | | Sham stimulation | 68. 6 ±3.2 | 11/29 | 40 | Neiguan (PC6), Hegu (LI4) | From 30 minutes pre-surgery to its conclusion | None |
| Duan et al. | 2019 | China | Randomized controlled trial | (1)(3) | TEAS | 78±10 | NA | 40 | Neiguan (PC6), Hegu (LI4) | From 30 minutes pre-surgery to its conclusion | 2–200 Hz |
| | | | | | Sham stimulation | 76±11 | NA | 40 | Neiguan (PC6), Hegu (LI4) | From 30 minutes pre-surgery to its conclusion | None |
| Yin et al. | 2015 | China | Randomized controlled trial | (3)(5) | TEAS | 78.3 ±5.5 | 17/5 | 27 | Baihui (GV 20), Neiguan (PC 6), and Fengchi(GB 20) | From 30 minutes pre-surgery to its conclusion | 2–200 Hz |
| | | | | | Sham stimulation | 77.5 ±5.2 | 16/5 | 26 | Baihui (GV 20), Neiguan (PC 6), and Fengchi(GB 20) | From 30 minutes pre-surgery to its conclusion | None |
| Lu et al. | 2019 | China | Randomized controlled trial | (1)(2)(3) (5) | TEAS | 72.07 ±2.53 | 22/24 | 46 | Baihui (GV 20), Neiguan (PC 6), and Fengchi(GB 20) | From 30 minutes pre-surgery to its conclusion | 2–100 Hz |
| | | | | | No stimulation | 71.29 ±2.31 | 24/21 | 45 | Baihui (GV 20), Neiguan (PC 6), and Fengchi(GB 20) | From 30 minutes pre-surgery to its conclusion | None |
| Liu et al. | 2019 | China | Randomized controlled trial | (4) | TEAS | 70.6 ±5.8 | 11/9 | 30 | Yuyao (EX-HN4), and Fengchi(GB 20) | From 30 minutes pre-surgery to its conclusion | 2–100 Hz |
| | | | | | Sham stimulation | 70.5 ±5.7 | 9/21 | 30 | Yuyao (EX-HN4), and Fengchi(GB 20) | From 30 minutes pre-surgery to its conclusion | None |
| Peng ta al. | 2019 | China | Randomized controlled trial | (1)(2)(6) (7)(8) | TEAS | 69.1 ±3.1 | 18/22 | 40 | Neiguan (PC6), Hegu (LI4) | From 30 minutes pre-surgery to 30 minutes after surgery | 2–100 Hz |
| | | | | | Sham stimulation | 68.7 ±3.3 | 13/27 | 40 | Neiguan (PC6), Hegu (LI4) | From 30 minutes pre-surgery to 30 minutes after surgery | None |

(*Continued*)

**Table 2.** (Continued)

| Author | Year | Country | Study design | Outcome | Comparison | Mean age±SD | Male/Female | Number of patients | Locations of electrodes placements | Duration of operation | Frequency |
|---|---|---|---|---|---|---|---|---|---|---|---|
| Wang DD et al. | 2016 | China | Randomized controlled trial | (1)(3)(4)(5) | TEAS | 69.9 ±4.2 | 16/14 | 30 | Baihui (GV20), Neiguan (PC6), Zusanli (ST36), Sanyinjiao (SP6) | From the begging of surgery to the end of surgery | 2–100 Hz |
| | | | | | No stimulation | 69.3 ±4.1 | 20/10 | 30 | None | None | None |
| Wang JW et al. | 2017 | China | Randomized controlled trial | (1)(2)(9) | TEAS | 70.23 ±5.51 | 14/17 | 31 | Huantiao(GB30), Fengshi(GB31), and Zusanli(ST36) | 30 minutes | 2–10 Hz |
| | | | | | Sham stimulation | 68.94 ±6.12 | 15/16 | 31 | Huantiao(GB30), Fengshi(GB31), and Zusanli(ST36) | 30 minutes | None |
| Ge et al. | 2023 | China | Randomized controlled trial | (1)(2) | TEAS | 63.2 ±1.29 | 15/15 | 30 | Neiguan (PC6), Hegu (LI4), Fengchi (GB20), and Zusanli (ST36) | 30 minutes | 2–100 Hz |
| | | | | | Sham stimulation | 62.0 ±1.20 | 13/17 | 30 | Neiguan (PC6), Hegu (LI4), Fengchi (GB20), and Zusanli (ST36) | 30 minutes | None |
| Li et al. | 2020 | China | Randomized controlled trial | (1)(2) | TEAS | 64.5 ±14.8 | 22/28 | 50 | Around the surgical incision | 40 minutes | 100 Hz |
| | | | | | Sham stimulation | 62.1 ±13.1 | 27/23 | 50 | NA | 40 minutes | None |

NA not available; (1) Postoperative Day 1 Visual Analog Scale; (2) Postoperative Day 2 Visual Analog Scale; (3) Incidence of Postoperative Cognitive Dysfunction; (4) Postoperative Day 1 Mini-Mental State Examination; (5) Postoperative Day 3 Mini-Mental State Examination; (6) Incidence of Nausea; (7) Incidence of Vomiting; (8) Incidence of Pruritus; (9) Incidence of Dizziness.

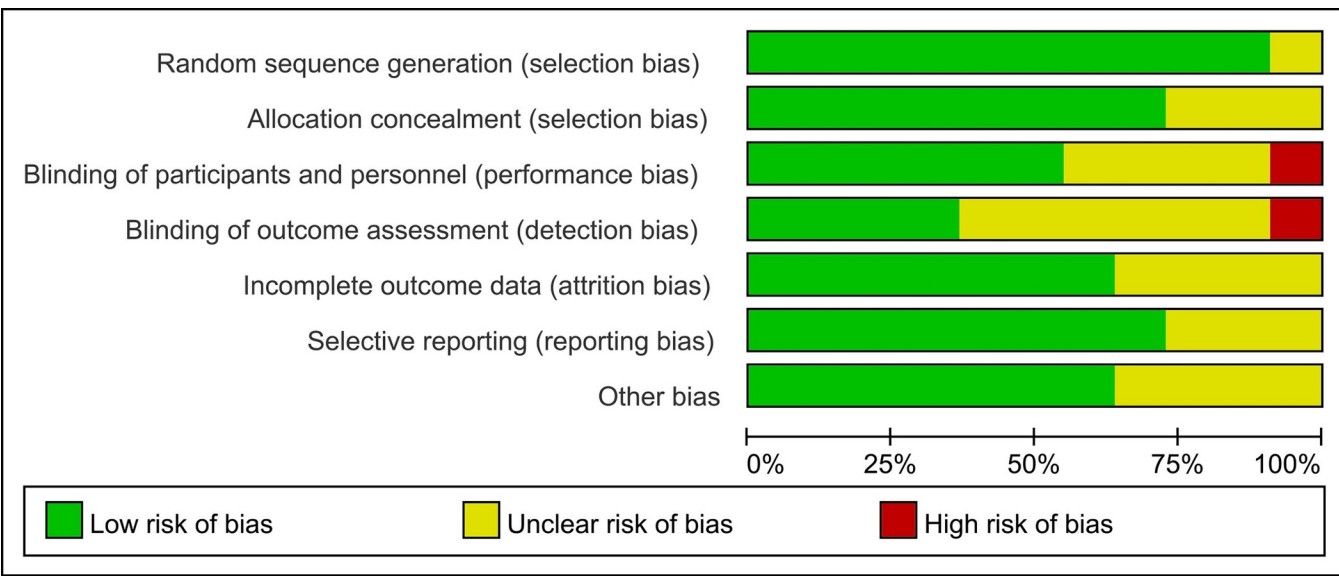

**Fig 2. Assessment of study quality using the Cochrane Risk of Bias Tool.** A color-coded evaluation of each domain (low risk. green, unclear risk. yellow, high risk. red).

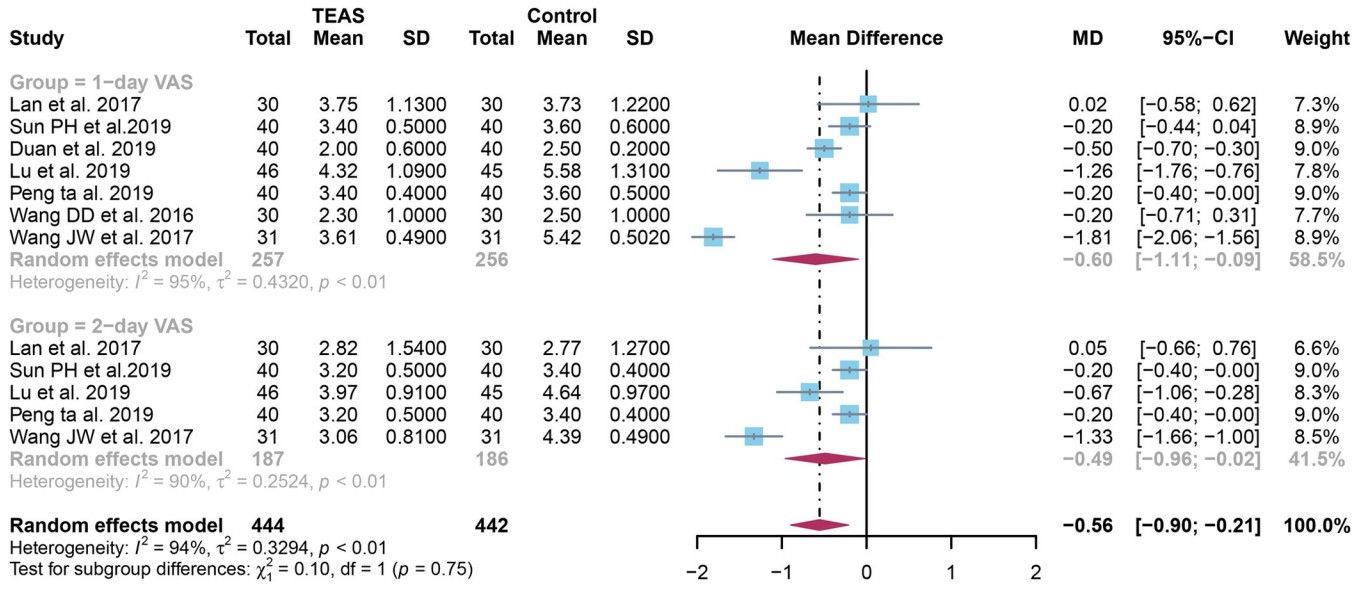

**Fig 3. Forest plot comparison of 1-day and 2-day VAS scores between TEAS and control groups.** Detailing individual study effect sizes (squares) with 95% confidence intervals (horizontal lines) and the aggregate pooled effect size (diamond) from all studies. VAS, visual analog scale; TEAS, transcutaneous electrical acupoint stimulation.

assessments. Despite these highlighted areas of potential concern, the overall risk of bias across the studies was considered to be within acceptable limits.

### 3.3 Quantitative results of 1-day and 2-day VAS scores

For 1-day VAS scores, a total of 9 studies were analyzed and random effect models were applied as significant heterogeneity($I^2 = 90\%$ and $P<0.01$). The 1-day VAS scores were significantly lower in the TEAS group compared to the control group (SMD: -0.78, 95% CI: -1.47, -0.09, $P = 0.02$) (Fig 3).

For 2-day VAS scores, a total of 7 studies were analyzed and random effect models were applied as significant heterogeneity($I^2 = 80\%$ and $P<0.01$).The 2-day VAS scores were significantly lower in the TEAS group compared to the control group (SMD:-0.54, 95% CI:-1.00,-0.09,$P = 0.02$) (Fig 3).

### 3.4 Quantitative results of 1-day and 3-day MMSE scores

For 1-day MMSE scores, a total of 3 studies were analyzed and random effect models were applied as low heterogeneity($I^2 = 83\%$ and $P<0.01$). The 1-day MMSE scores were significantly higher in the TEAS group compared to the control group (SMD: 1.60, 95% CI: 0.68, 2.51,$P<0.001$) (Fig 4).

For 3-day MMSE scores, a total of 4 studies were analyzed and fixed effect models were applied as low heterogeneity($I^2 = 0\%$ and $P = 0.41$). The 3-day MMSE scores were significantly lower in the TEAS group compared to the control group (SMD:1.31, 95% CI:1.03,1.59, $P<0.001$) (Fig 4).

### 3.5 Quantitative results of POCD rate

A total of 6 studies were analyzed and fixed effect models were applied as no significant heterogeneity ($I^2 = 0\%$ and $P = 0.74$). The POCD rate was significantly lower in the TEAS group compared to the control group (RR: 0.55, 95% CI: 0.41, 0.73, $P<0.001$) (Fig 5).

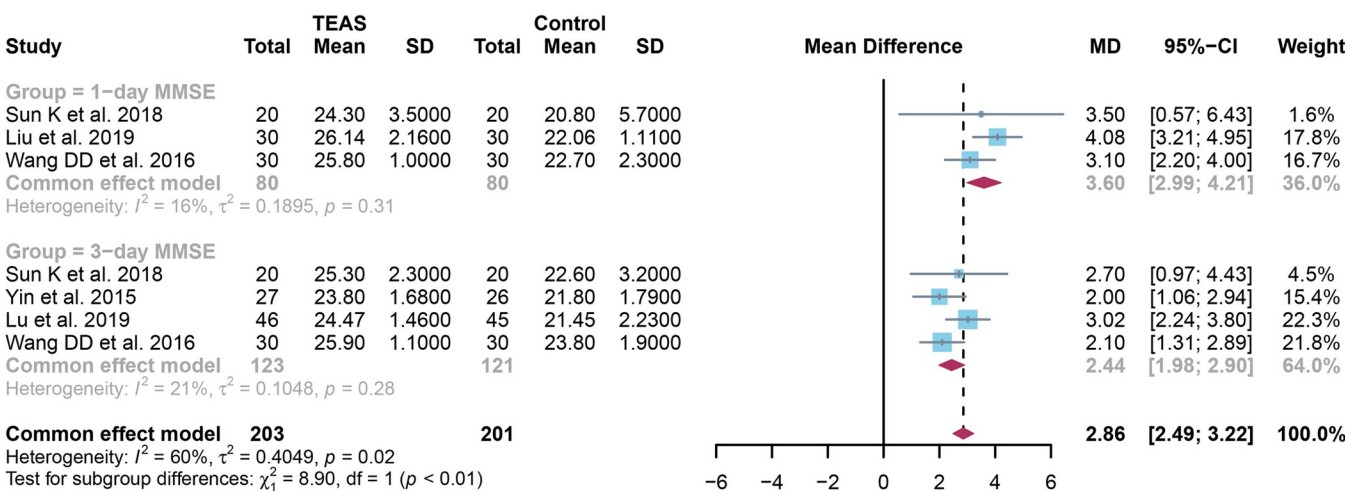

**Fig 4. Forest plot comparison of 1-day and 3-day MMSE scores between TEAS and control groups.** Detailing individual study effect sizes (squares) with 95% confidence intervals (horizontal lines) and the aggregate pooled effect size (diamond) from all studies. MMSE, mini-mental state examination; TEAS, transcutaneous electrical acupoint stimulation.

### 3.6 Quantitative results of adverse event rate

For adverse event rate, fixed effect models were employed due to the absence of significant heterogeneity ($I^2 = 0\%$, $P = 0.97$). The findings reveal a significantly lower incidence rate of nausea in the TEAS group compared to the control group (RR: 0.41, 95% CI: 0.24, 0.69, $P<0.001$). Similarly, the rate of vomiting was significantly reduced in the TEAS group (RR: 0.32, 95% CI: 0.17, 0.58 $P<0.001$). No significant difference was observed in the occurrence of pruritus between the two groups (RR: 0.75, 95% CI: 0.41, 1.39 $P = 0.72$). However, the incidence of dizziness was significantly lower in the TEAS group (RR: 0.36, 95% CI: 0.17, 0.76, $P<0.001$) (Fig 6).

## 4. Discussion

Recent studies have shown TEAS to be effective in enhancing cognitive functions and alleviating pain post-hip replacement surgery [24, 28], yet some studies report contradictory findings [20, 27]. The body of research on TENS and its clinical efficacy is substantial, but challenges in interpreting these studies arise from insufficient design and methodology reporting, alongside

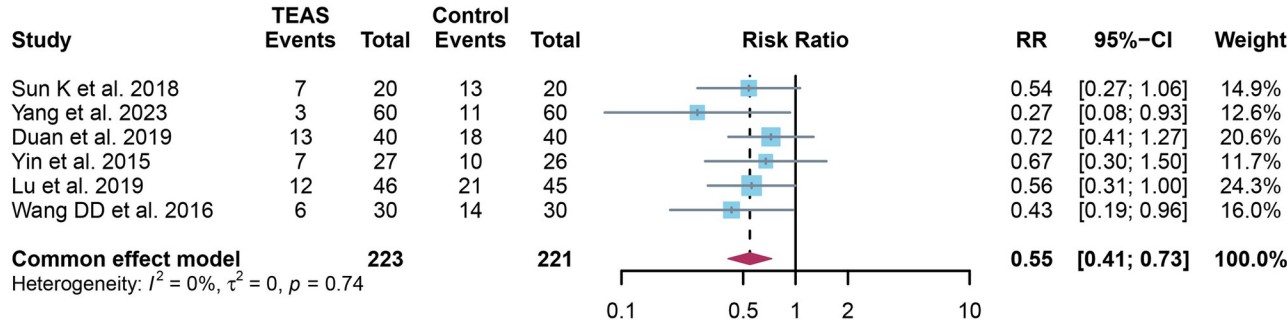

**Fig 5. Forest plot comparison of POCD rate between TEAS and control groups.** Detailing individual study effect sizes (squares) with 95% confidence intervals (horizontal lines) and the aggregate pooled effect size (diamond) from all studies. POCD, postoperative cognitive dysfunction; TEAS, transcutaneous electrical acupoint stimulation.

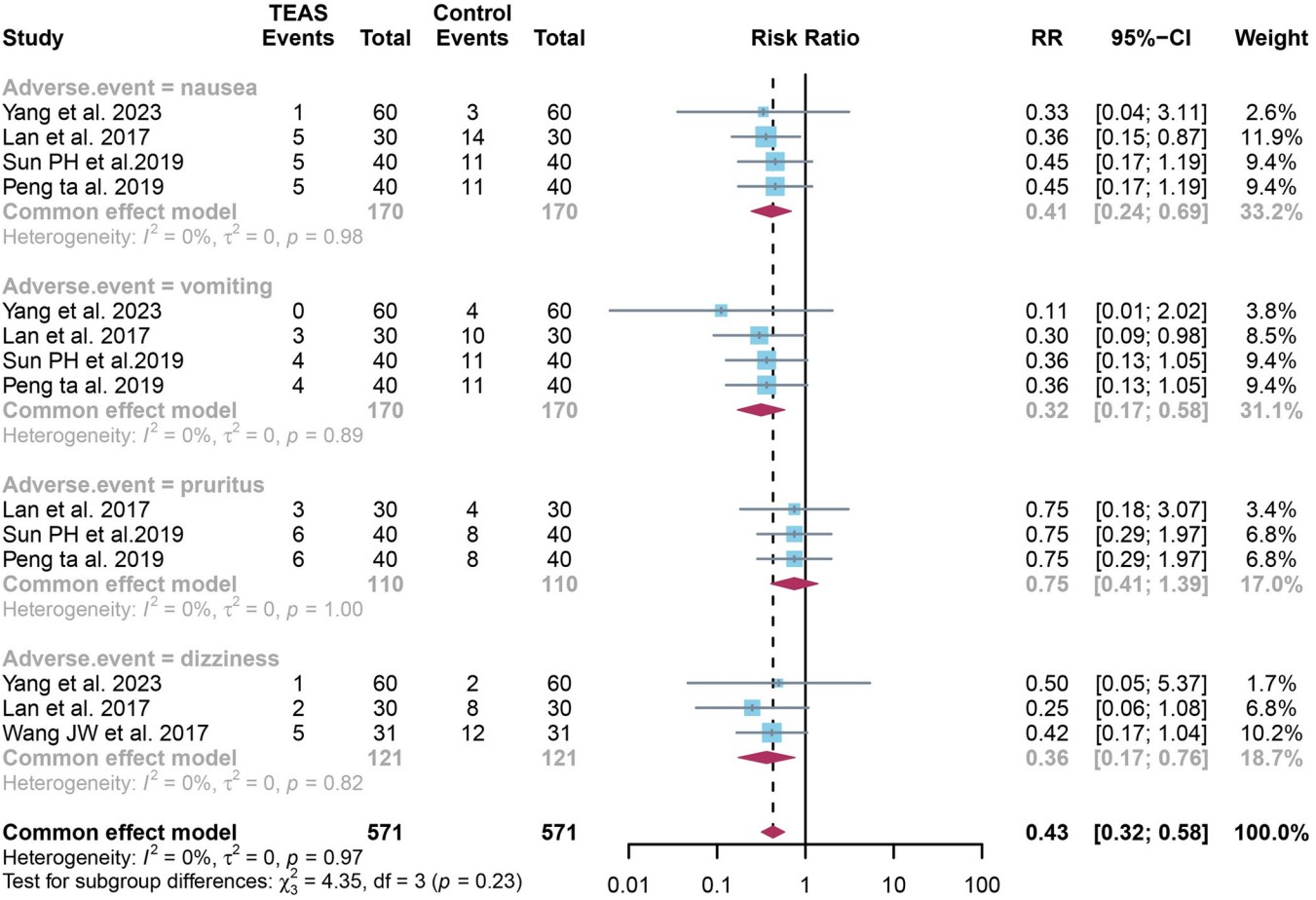

**Fig 6. Forest plot comparison of adverse event (nausea, vomiting, dizziness, and pruritus) rate between TEAS and control groups.** Detailing individual study effect sizes (squares) with 95% confidence intervals (horizontal lines) and the aggregate pooled effect size (diamond) from all studies. TEAS, transcutaneous electrical acupoint stimulation.

concerns regarding TENS technique application [18, 21, 22]. The specific efficacy and safety of TEAS remain to be clarified. Therefore, we performed the first meta-analysis focusing on the impact of TEAS on cognitive functions and pain management for elderly patients around the perioperative period of hip replacement surgery.

This meta-analysis revealed that TEAS significantly reduced pain after hip replacement surgery, as shown by lower 1-day and 2-day VAS scores in the TEAS group compared to controls. The mechanistic basis for this improvement may be attributed to TEAS's ability to modulate pain pathways and neurotransmitter release, thus enhancing the threshold for pain perception and facilitating postoperative pain management, suggesting its potential as a non-pharmacological pain management option [31–33].

Additionally, TEAS was associated with improved cognitive function, evidenced by higher MMSE scores and a reduced rate of POCD. The mechanism behind TEAS's enhancement of cognitive outcomes post-surgery may involve its stimulation of cerebral blood flow and modulation of the autonomic nervous system, potentially mitigating the cognitive impairments often associated with surgical stress and anesthesia [24, 34, 35]. Moreover, TEAS effectively reduced nausea, vomiting, and dizziness, highlighting its influence on the autonomic and vestibular systems. This reduction in postoperative adverse effects suggests TEAS enhances

elderly patient comfort and may accelerate recovery, underscoring its value in post-surgical care.

In 2017, Johnson highlighted the potential of TEAS as an adjunctive tool in perioperative pain management [10]. Johnson's critical review emphasized TEAS's benefits in reducing analgesic consumption, improving pain, enhancing pulmonary function, and alleviating nausea and vomiting [10]. While Johnson's review leaned towards supporting TENS's effectiveness over placebo in relieving acute pain and reducing analgesic consumption, our study advances this field by utilizing meta-analytical statistical techniques to further validate this hypothesis.

On the other hand, Davison et al. (2021) conducted a systematic review to assess the effectiveness of electrical stimulation in promoting clinical outcomes post-hip fractures [36]. Their inclusion of only four studies revealed that electrical stimulation, notably TEAS, not only reduced pain (MD = 3.3 points on VAS, $P < 0.001$) but also facilitated immediate functional recovery after a hip fracture ($P < 0.001$) [36]. Despite these promising findings, the review by Davison et al. suffers from a significant limitation in its scope and depth, primarily due to the inclusion of a small number of studies and a lack of comprehensive evidence. Moreover, their review did not address improvements in cognitive function or evaluate the potential adverse reactions associated with TEAS. These limitations called for the current meta-analysis.

In evaluating TEAS as a modality for improving pain and cognitive function in patients around the perioperative period of hip replacement surgery, it highlights significant advantages, including cost-effectiveness, reduced reliance on pain medication, and potentially shorter hospital stays [10, 36]. These benefits underscore the appeal of TEAS as a non-pharmacological intervention that aligns with current trends towards minimizing opioid use and enhancing elderly patient recovery processes. However, the difference in stimulation parameters and the duration of interventions introduce a degree of heterogeneity to the studies reviewed, complicating the establishment of a universally optimal TEAS protocol. Preliminary evidence suggests the efficacy of a more isolated and patient-comfort-focused lower-frequency protocol for individuals post-hip fracture, however, there is a need for further research to delineate the optimal parameters for TEAS application [37, 38]. Future studies not only explore the cost-effectiveness of TEAS in comparison to traditional pain management methods, such as medications but also aim to establish standardized protocols that maximize the benefits of TEAS.

Some limitations of the current meta-analysis should be considered when interpreting the results. Firstly, the heterogeneity of the included studies may have affected the overall 1-day VAS and 2-day VAS, possibly due to differences in TEAS techniques (such as varying parameters and stimulation durations) among the studies. Secondly, the studies included in the meta-analysis were relatively small, especially those evaluating cognitive function and safety, which may have introduced bias. Therefore, well-designed prospective studies with larger sample sizes are needed to confirm the findings of this meta-analysis.

## 5. Conclusion

Our meta-analysis demonstrated that TEAS significantly reduces pain and improves cognitive function in elderly patients undergoing hip replacement surgery. Future studies should further investigate the optimal TEAS protocols to maximize these benefits across different population and surgical settings.

## Supporting information

**S1 Checklist. PRISMA 2020 checklist.**
(DOCX)

**S1 File.**
(DOCX)

**S2 File.**
(DOCX)

**S1 Data.**
(XLSX)

## Author Contributions

**Data curation:** Sujuan Xu, Kai Huang, Qing Jiang.

**Formal analysis:** Sujuan Xu, Qing Jiang.

**Project administration:** Sujuan Xu, Qing Jiang.

**Software:** Qing Jiang.

**Validation:** Qing Jiang.

**Visualization:** Sujuan Xu.

**Writing – original draft:** Sujuan Xu, Kai Huang.

**Writing – review & editing:** Qing Jiang.

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
