## [Decision Letter · Decision Letter 0]

30 Jul 2024

PONE-D-24-21132Evaluation of Transcutaneous Electrical Acupoint Stimulation for Improving Pain and Cognitive Function in Patients Around the Perioperative Period of Hip Replacement Surgery: A Meta-AnalysisPLOS ONE

Dear Dr. Jiang,

Thank you for submitting your manuscript to PLOS ONE. After careful consideration, we feel that it has merit but does not fully meet PLOS ONE’s publication criteria as it currently stands. Therefore, we invite you to submit a revised version of the manuscript that addresses the points raised during the review process.

We look forward to receiving your revised manuscript.

Kind regards,

Hantong Hu

Academic Editor

PLOS ONE

 [Zhejiang Province Traditional Chinese Medicine Science and Technology Plan Project(NO.2024ZL029)].  

3. PLOS requires an ORCID iD for the corresponding author in Editorial Manager on papers submitted after December 6th, 2016. Please ensure that you have an ORCID iD and that it is validated in Editorial Manager. To do this, go to ‘Update my Information’ (in the upper left-hand corner of the main menu), and click on the Fetch/Validate link next to the ORCID field. This will take you to the ORCID site and allow you to create a new iD or authenticate a pre-existing iD in Editorial Manager. Please see the following video for instructions on linking an ORCID iD to your Editorial Manager account: https://www.youtube.com/watch?v=_xcclfuvtxQ.

Additional Editor Comments (if provided):

Reviewers' comments:

Reviewer's Responses to Questions

**Comments to the Author**

1. Is the manuscript technically sound, and do the data support the conclusions?

Reviewer #1: Yes

Reviewer #2: Yes

2. Has the statistical analysis been performed appropriately and rigorously? 

Reviewer #1: N/A

Reviewer #2: No

3. Have the authors made all data underlying the findings in their manuscript fully available?

Reviewer #1: Yes

Reviewer #2: Yes

4. Is the manuscript presented in an intelligible fashion and written in standard English?

Reviewer #1: Yes

Reviewer #2: Yes

5. Review Comments to the Author

Reviewer #1: The manuscript with title "Evaluation of Transcutaneous Electrical Acupoint Stimulation for Improving

Pain and Cognitive Function in Patients Around the Perioperative Period of Hip

Replacement Surgery: A Meta-Analysis" was good written and designed

Reviewer #2: In this manuscript, the research team employed meta-analysis and systematic evaluation to investigate the efficacy and safety of transcutaneous electrical acupoint stimulation in patients around the perioperative period of hip replacement surger. The author's discussion is clearly expressed, allowing readers to easily comprehend the study's theme. However, this manuscript still requires some improvements. Here are my suggestions.

1. Due to the small number of documents included in this study, it is recommended to consider expanding the age range of the subjects. If elderly patients undergoing hip replacement surgery are selected, please describe the purpose of the study in the title and the full text.

2. Based on Table 2, it is evident that China is predominantly involved in research within this field. Therefore, it is essential to search four renowned databases in China to ensure comprehensive retrieval of literature. Additionally, please update the retrieval date.

3. The author should consider whether there are different measurement methods or units between VAS and MMSE, which will bring heterogeneity. Therefore, is it more accurate to utilize SMD in the scoring table?

4. After evaluating the risk bias of the included literature, two graphs are usually generated. Please fill in the missing detail evaluation chart so that readers can understand the risk of bias in each included study.

5. Publication bias is usually carried out when an outcome indicator meets the number of 10 or more studies.

6. In the results section, please add the P value of each comparison result.

6. PLOS authors have the option to publish the peer review history of their article (what does this mean?). If published, this will include your full peer review and any attached files.

Reviewer #1: No

Reviewer #2: No

---

## [Author Response · Author response to Decision Letter 0]

14 Aug 2024

List of Responses

Dear Editor and Reviewers: 

Thank you for your letter and for the reviewers’ comments concerning our manuscript entitled “Evaluation of Transcutaneous Electrical Acupoint Stimulation for Improving Pain and Cognitive Function in Patients Around the Perioperative Period of Hip Replacement Surgery: A Meta-Analysis”. ID:PONE-D-24-21132. Those comments are all valuable and very helpful for revising and improving our paper, as well as the important guiding significance to our research. We have studied comments carefully and have made correction which we hope meet with approval. Revised portion are marked in red in the revised paper. The main corrections in the paper and the responds to the reviewer’s comments are as flowing:

Responds to the reviewer’s comments:

Reviewer #1:

Comment 1: The manuscript with title "Evaluation of Transcutaneous Electrical Acupoint Stimulation for Improving Pain and Cognitive Function in Patients Around the Perioperative Period of Hip Replacement Surgery: A Meta-Analysis" was good written and designed.

Answer: Thank you for your approval.

Reviewer #2: 

Comment 1: In this manuscript, the research team employed meta-analysis and systematic evaluation to investigate the efficacy and safety of transcutaneous electrical acupoint stimulation in patients around the perioperative period of hip replacement surgery. The author's discussion is clearly expressed, allowing readers to easily comprehend the study's theme.

Answer: Thank you for your approval.

Comment 2: Due to the small number of documents included in this study, it is recommended to consider expanding the age range of the subjects. If elderly patients undergoing hip replacement surgery are selected, please describe the purpose of the study in the title and the full text.

Answer: We would like to express our sincere gratitude for your valuable feedback and insightful comments.

Regarding your suggestion to expand the age range of the subjects, we appreciate your concern. However, the primary population undergoing hip replacement surgery is indeed elderly patients. Our study was specifically aimed at providing clinical guidance for this patient group. We apologize for not emphasizing this focus on elderly patients earlier in the manuscript.

In response to your suggestion, we have made the necessary revisions. We have updated the title to "Evaluation of Transcutaneous Electrical Acupoint Stimulation for Improving Pain and Cognitive Function in Elderly Patients Around the Perioperative Period of Hip Replacement Surgery: A Meta-Analysis" to better reflect the targeted population. Additionally, we have emphasized the focus on elderly patients throughout the full text, including in the purpose.

All changes have been highlighted in red in the revised manuscript for your convenience

Comment 3: Based on Table 2, it is evident that China is predominantly involved in research within this field. Therefore, it is essential to search four renowned databases in China to ensure comprehensive retrieval of literature. Additionally, please update the retrieval date.

Answer: We sincerely appreciate your insightful comments and valuable suggestions.

In response to your recommendation, we have conducted a more thorough literature search by adding two additional Chinese databases, the VIP Database and SinoMed. We have re-screened all four Chinese databases to ensure comprehensive retrieval of relevant literature. The complete search strategies have been included in Table 1 for your reference.

As a result of this extended search, we identified two new studies, Ge et al.[1] and Li et al.[2], which have been included in our analysis. We have updated the data, figures, tables, and the retrieval date accordingly.

1. Ge Y. Observation of the Analgesic Effect of Transcutaneous Acupoint Electrical Stimulation During the Perioperative Period of Total Hip Arthroplasty(in Chinese). Zhejiang Journal of Traditional Chinese Medicine. 2023;58(6):439. doi: 10.13633/j.cnki.zjtcm.2023.06.010.

2. Li W, Wang B, Wang K, Xu M, Yin H, He X. Effect of transcutaneous electrical nerve stimulation on pain after total hip arthroplasty(in Chinese). Chin J Bone Joint Surg. 2020;13(3). doi: 10.3969/j.issn.2095-9958.2020.03.13.

Comment 4: The author should consider whether there are different measurement methods or units between VAS and MMSE, which will bring heterogeneity. Therefore, is it more accurate to utilize SMD in the scoring table?

Answer: Thank you for your thoughtful feedback and for raising an important point regarding potential heterogeneity due to different measurement methods or units between VAS and MMSE.

We agree that this could indeed introduce heterogeneity in our analysis. Following your suggestion, we have utilized the Standardized Mean Difference (SMD) for the following four indicators: (1) Postoperative Day 1 Visual Analog Scale, (2) Postoperative Day 2 Visual Analog Scale, (3) Postoperative Day 1 Mini-Mental State Examination, and (4) Postoperative Day 3 Mini-Mental State Examination. We have updated the forest plots accordingly.(Please See Fig 3 and Fig 4)

Comment 5: After evaluating the risk bias of the included literature, two graphs are usually generated. Please fill in the missing detail evaluation chart so that readers can understand the risk of bias in each included study.

Answer: We sincerely appreciate your careful evaluation and constructive suggestion.

In response to your comment, we have replaced the previous quality assessment figures with detailed evaluation charts. These charts now clearly present the risk of bias for each included study, including the two newly added articles. This will allow readers to better understand the risk of bias in each study.(Please See Fig 2)

Comment 6: Publication bias is usually carried out when an outcome indicator meets the number of 10 or more studies.

Answer: Thank you for your insightful comment regarding the assessment of publication bias.

After reviewing the Cochrane Handbook, we agree with your observation. As a result, we have removed the publication bias assessment and the associated funnel plots from our analysis to avoid presenting potentially misleading or non-objective results.

Comment 7: In the results section, please add the P value of each comparison result.

Answer: Thank you for your attention to detail and for your valuable feedback.

In response to your suggestion, we have added the P values for each comparison result in the results section of the manuscript. These additions will provide a clearer understanding of the statistical significance of our findings.

We tried our best to improve the manuscript and made some changes in the manuscript. These changes will not influence the content and framework of the paper. And all the changes have mark in red in the revised paper. We appreciate for Editors/Reviewers’ warm work earnestly, and hope that the correction will meet with approval.

Once again, thank you very much for your comments and suggestions.

---

## [Editor Report · Decision Letter 1]

16 Aug 2024

Evaluation of Transcutaneous Electrical Acupoint Stimulation for Improving Pain and Cognitive Function in Elderly Patients Around the Perioperative Period of Hip Replacement Surgery: A Meta-Analysis

PONE-D-24-21132R1

Dear Dr. Jiang

We’re pleased to inform you that your manuscript has been judged scientifically suitable for publication and will be formally accepted for publication once it meets all outstanding technical requirements.

Kind regards,

Hantong Hu

Academic Editor

PLOS ONE

Additional Editor Comments (optional):

The authors have made substantial revisions to the article strictly based on the reviewers’ comments, significantly improving the quality of the paper. I believe it now meets the standards for acceptance.
---

## [Editor Report · Acceptance letter]

28 Aug 2024

PONE-D-24-21132R1 

PLOS ONE

Dear Dr. Jiang, 

I'm pleased to inform you that your manuscript has been deemed suitable for publication in PLOS ONE. Congratulations! Your manuscript is now being handed over to our production team.

Kind regards, 

on behalf of

Dr. Hantong Hu 

Academic Editor

PLOS ONE